



# The Lagrangian Atmospheric Radionuclide Transport Model (ARTM) - Sensitivity studies and evaluation using airborne measurements of power plant emissions

Robert Hanfland[1,2], Dominik Brunner[3], Christiane Voigt[2,4], Alina Fiehn[4], Anke Roiger[4], and
Margit Pattantyús-Ábrahám[1]

[1]Bundesamt für Strahlenschutz, Oberschleißheim, Germany
[2]Johannes Gutenberg-Universität Mainz, Institute of Atmospheric Physics, Mainz, Germany
[3]Empa Swiss Federal Laboratories for Material Science and Technology, Dübendorf, Switzerland
[4]Deutsches Zentrum für Luft- und Raumfahrt (DLR), Institute of Atmospheric Physics, Oberpfaffenhofen, Germany
**Correspondence:** Robert Hanfland (rhanfland@bfs.de)

**Abstract.** The Atmospheric Radionuclide Transport Model (ARTM) operates at the meso-$\gamma$-scale and simulates the dispersion of radionuclides originating from nuclear facilities under routine operation within the planetary boundary layer. This study presents the extension and validation of this Lagrangian particle dispersion model and consists of three parts: i) a sensitivity study that aims to assess the impact of key input parameters on the simulation results; ii) the evaluation of the mixing prop-

erties of five different turbulence models using the well-mixed criterion; and iii) a comparison of model results to airborne observations of carbon dioxide ($CO_2$) emissions from a power plant and the evaluation of related uncertainties. In the sensitivity study, we analyse the effects of stability class, roughness length, zero-plane displacement factor and source height on the three-dimensional plume extent as well as the distance between source and maximum concentration at the ground. The results show that the stability class is the most sensitive input parameter as expected. The five turbulence models are the default turbu-

lence models of ARTM 2.8.0 and ARTM 3.0.0, one alternative built-in turbulence model of ARTM and two further turbulence models implemented for this study. The well-mixed condition tests showed that all five turbulence models are able to preserve an initially well-mixed atmospheric boundary layer reasonably well. The models deviate only 6% from the expected uniform concentration below $80\%$ of the mixing layer height except for the default turbulence model of ARTM 3.0.0 with deviations by up to $18\%$, respectively. $CO_2$ observations along a flight path in the vicinity of the lignite power plant Bełchatów, Poland

measured by the DLR Cessna aircraft during the CoMet campaign in 2018 allow to evaluate the model performance for the different turbulence models under unstable boundary layer conditions. All simulated mixing ratios are in the same order of magnitude as the airborne in situ data. An extensive uncertainty analysis using probability distribution functions, statistical tests and direct spatio-temporal comparisons of measurements and model results help to quantify the model uncertainties. With the default turbulence setups of ARTM version 2.8.0 and 3.0.0, the plume widths are underestimated by up to $50\%$ resulting in

a strong overestimation of the maximum plume $CO_2$ mixing ratios. The comparison of the three alternative turbulence models shows a good agreement of the peak plume $CO_2$ concentrations, the $CO_2$ distribution within the plumes and the plume width with 30% deviations in peak $CO_2$ concentration and less than $25\%$ deviation of the measured $CO_2$ plume width. Uncertainties





of the simulations may arise from the different spatial and temporal resolution of simulations and measurements in addition to the turbulence parametrisation and boundary conditions. The results of this work may help to improve the accurate representa-
tion of real plumes in very unstable atmospheric conditions by the selection of distinct turbulence models. Further comparisons at different stability regimes are required for a final assessment of model uncertainties.

## 1 Introduction

Atmospheric dispersion models (ADMs) are widely used by the scientific community and authorities. They are applied to a
variety of problems, such as the study of the impact of pollutant emissions on air quality (Gariazzo et al., 2007; Stohl et al., 2007; Berchet et al., 2017; Lonati et al., 2022; Shupe et al., 2022) or the dispersion of radioactive discharges to the air (Chino et al., 2011; Connan et al., 2013; Draxler et al., 2015; Arnold et al., 2015) and they can operate at the full meteorological scale ranging from the micro-scale (shorter than $\mathrm{km}$) to the meso-scale ($\mathrm{km}$ to thousand $\mathrm{km}$) up to the synoptic or global scale (larger than thousand $\mathrm{km}$).

The Atmospheric Radionuclide Transport Model (ARTM), analysed in this study, belongs to the class of models operating at the micro-$\beta$ to meso-$\gamma$-scale (approx. $0.5\,\mathrm{km}$ to $20\,\mathrm{km}$). It is a Lagrangian particle dispersion model (LPDM) designed for the dispersion of radionuclides from nuclear facilities under routine operation in the planetary boundary layer (PBL).

However, any ADM has to demonstrate the applicability to the system of study. The most important method to confirm this is validation (Kleijnen, 1995; Schlesinger et al., 1979). This includes i) sensitivity analysis (SA), which relates the model
response to variations of the model's input parameters as well as ii) the comparison with observations revealing whether a model is an accurate representation of the system and whether simulation results are, to a certain degree, in agreement with observations (Kleijnen, 1995; Rao, 2005).

Concerning ARTM, Hettrich (2017) performed a sensitivity study to analyse the effect of input parameters (e.g. emission rate, source geometry, stability class and particle number) on concentrations at selected locations near the ground. Hanfland
et al. (2022) provided an overview of the physical basis and mathematical formulations of the model and presented a qualitative description of the influences of different input parameters on three-dimensional plume characteristics for a general simulation setup. However, both lack in quantifying sensitivities.

Here we expand the former studies with a more systematic and quantitative sensitivity analysis. Different sensitivity coefficients are calculated, which describe the dependence of the simulation output on the input parameters stability class (SC),
roughness length ($z_0$), zero-plane displacement factor ($d$) and source height ($h_s$) within the whole simulated PBL and rank them according to their effects on the model output.

Publications presenting comparisons of ARTM's mixing ratio simulation results with measurements are rare. Hettrich (2017) compared ARTM simulation results with measurements at a few selected locations near the surface showing discrepancies





which are related to complex orography or local thermal induced winds that are not covered by ARTM's wind field model
TALdia. Martens et al. (2012) studied the influence of a single large building close to the source on the dispersion showing that
at distances larger than $4\,\mathrm{km}$ the influence decreases. These comparisons covered only a few atmospheric conditions and were
limited to near surface concentration measurements.

Brunner et al. (2022) presented an intercomparison of six different atmospheric transport models, including ARTM, with
airborne in situ and remote sensing carbon dioxide ($CO_2$) measurements sampling the exhaust plume of the Bełchatów lignite
power plant in Poland under very unstable atmospheric conditions (Fix et al., 2018). The data set comprises a considerable
number of plume transects at different distances from the source and heights within the PBL, is characterised by a strong
contrast between background and plume $CO_2$ mixing ratio and provides a three-dimensional description of the mixing ratio
field. The spatial extent of the area covered by measurements is around the maximum domain size ARTM can tackle. In that
study ARTM simulations were performed using the default turbulence model of version 2.8.0. including a workaround for
meandering plumes because the simulated plume appeared to be too narrow at very unstable atmospheric conditions.

ARTM's dispersion depends on the used turbulence model (Hanfland et al., 2022) and since the three-dimensional data set
facilitates further analysis of the model we investigate whether the modelled plume could become more realistic by using
different turbulence models. Two new turbulence models were implemented based on the ideas of Hanna (1982) and Degrazia
et al. (2000) additionally to three built-in models of ARTM. The same workaround for meandering plumes based on the default
turbulence model of ARTM 2.8.0, as primarily presented in Brunner et al. (2022), is also included to this study. Furthermore, all
the five turbulence models are evaluated concerning turbulence model characteristics and mixing efficiency as well as compared
with measurements.

The structure of this study is as follows: Section 2 gives a short description of the atmospheric dispersion model ARTM.
Section 3 introduces several different sensitivity analysis methods and presents the sensitivity of typical model simulation
outputs to key input parameters. Section 4 presents the five turbulence models and assesses their performance with respect to
the well-mixed condition. Section 5 shows the comparison and evaluation of ARTM simulation results for the five turbulence
models with the three-dimensional airborne data. Section 6 concludes the results of this paper.

## 2   The model ARTM

The Atmospheric Radionuclide Transport Model is a LPDM developed specifically for the dispersion of radioactive emissions
from nuclear facilities in an area of typically $10\,\mathrm{km} \times 10\,\mathrm{km}$. Its purpose is to provide annual activity concentration fields in
the area around nuclear facilities under routine operation in slightly structured non-urban terrain which are used in a follow-up
step to calculate the additional exposure of the population (Hanfland et al., 2022).

The dispersion model propagates numerical particles representing radioactive tracers in space and time according to wind
and turbulence fields obtained by a diagnostic approach. Meteorological data from measurements in the vicinity of the nuclear
facilities are used to calculate a mass-conserving diagnostic wind field. The turbulence is obtained by a Markov process which
uses wind speed fluctuations and Lagrangian correlation times as input parameters, both depending on the Obukhov length



as turbulence parameter. This static diagnostic approach employed by ARTM differs from other larger scale LPDMs such as FLEXPART, STILT, NAME or HYSPLIT, which use prognostic meteorological fields from a numerical weather prediction model to drive the particle propagation (Hanfland et al., 2022; Lin et al., 2003; Stohl et al., 2005; Ryall and Maryon, 1998;

Draxler and Hess, 1998). The advantage of this static approach is its high computational efficiency but its application is limited to the area in the vicinity of the meteorological measurement (Ratto et al., 1994; Eichhorn and Kniffka, 2010; Wan et al., 2013). The simulation domain is divided into grid cells for which the average activity concentration for the whole simulation period is calculated. A detailed model description is given in Hanfland et al. (2022).

## 3 Sensitivity study

Sensitivity analysis is a method to study the model's response to variations of input parameters in a systematic way and it may answer the following questions: how does the uncertainty of input parameters influence the model output; which parameters require additional research in order to reduce output uncertainty; which parameters are most significant or insignificant for the model's output; does the model behave as expected when varying a certain input parameter (Hamby, 1994; Frey and Patil, 2002; Rao, 2005; Saltelli et al., 2008). SA methods are either local or global depending on the sampled input parameter space

(Saltelli et al., 2008; Morio, 2011; Zagayevskiy and Deutsch, 2015).

The results of the methods may differ depending on the shape of the input parameter space. Thus, the application of several methods is recommended (Iman and Helton, 1988; Hamby, 1995). In this work, several different local and global SA methods are therefore applied to provide a comprehensive assessment of the response of ARTM to different input parameters.

### 3.1 Local sensitivity analysis methods

Local SA focuses on one single point in the input parameter space. The output of a model is represented by $Y = g(X_1, \ldots X_k)$ where the random variables $X_i$ with $i = 1, \ldots, k$ denote the different input parameters. The representations (or values) of $X_i$ are denoted with $x_i$. The input parameters $X_i$ are varied one at a time while all the others are held constant at their reference values $x_i^{\mathrm{ref}}$. This local SA approach is similar to estimating the partial derivative $\frac{\partial Y}{\partial X_i}$ and characterises the effect of the input parameter $X_i$ on $Y$ at one reference point $\boldsymbol{X}^{\mathrm{ref}} = (x_1^{\mathrm{ref}}, \ldots, x_k^{\mathrm{ref}})$ (Morio, 2011).

### 110 3.1.1 Sensitivity index

The sensitivity index described by Hoffman and Gardner (1983) uses the parameters at the reference point where each parameter is varied one at a time by their full range. The sensitivity index is calculated as

$$SI_i = \frac{Y_{i,\max} - Y_{i,\min}}{Y_{i,\max}} \tag{1}$$

where $Y_{i,\max(\min)}$ indicates the maximum (minimum) output value, respectively. The sensitivity index is a value between

$0 \leq SI_i \leq 1$ and gives the fraction of output variation caused by the varied input parameter (Hamby, 1994).



### 3.1.2 One-at-a-time sensitivity measure

The one-at-a-time sensitivity measure calculates the variation of the model output normalised to the largest output variation $\Delta Y_{\max}$ that had been observed for the different input parameters. Starting from the default parameter set, the parameters are varied one at a time by a percentage $\alpha$. The sensitivity coefficient for the input parameter $X_i$ are calculated as

$$SM_i^\alpha = \frac{|Y_{i,+\alpha} - Y_{i,-\alpha}|}{\Delta Y_{\max}} \tag{2}$$

where $\Delta Y_{\max} = \max(|Y_{l,+\alpha} - Y_{l,-\alpha}|) \; \forall l \in i$ (Link et al., 2018). In this work the percentages $\pm 25\,\%$ and $\pm 50\,\%$ are used for $\alpha$. $SM_i^\alpha$ is a value between $0 \leq SM_i^\alpha \leq 1$ where unity identifies the input parameter with the biggest effect on the model output $Y$.

## 3.2 Global sensitivity analyses

Global SAs sample the whole input parameter space, which leads to a broader representation of the sensitivity compared to local methods but also increase computation time. A general discussion about global SA can be found in Saltelli et al. (2008).

### 3.2.1 Sobol' indices

The variance-based Sobol' indices use variance decomposition to calculate indices of different orders (Sobol', 1993). Usually, only two key Sobol' indices are determined: the first-order index $S_i$; and the total effect index $S_{\mathrm{T}i}$.

For the first one, the conditional expected value of the model output $E_{\boldsymbol{X}_{\sim i}}(Y|X_i)$ with a constant value of $X_i$ and varying values for all other input parameters $\boldsymbol{X}_{\sim i}$ is computed. For different realisations of $X_i$, $V_{X_i}[E_{\boldsymbol{X}_{\sim i}}(Y|X_i)]$ reflects the variance of the model output $Y$ originating from a variation of the input parameter $X_i$. The first Sobol' index is then given by

$$S_i = \frac{V_{X_i}[E_{\boldsymbol{X}_{\sim i}}(Y|X_i)]}{V(Y)} \tag{3}$$

where $V(Y)$ is the unconditional variance of the output where all $X_i$ are varied. $V_{X_i}[E_{\boldsymbol{X}_{\sim i}}(Y|X_i)]$ cannot be larger than $V(Y)$ and thus for the sensitivity coefficient $0 \leq S_i \leq 1$ is valid. This index is called first-order sensitivity index as it does not take higher-order effects (i.e. interactions between different input parameters) into account (Saltelli et al., 2008).

The second index considered here is the total effect index. It takes higher order terms into account, which might be important depending on the model. The total effect is calculated as

$$S_{\mathrm{T}i} = 1 - \frac{V_{\boldsymbol{X}_{\sim i}}[E_{X_i}(Y|\boldsymbol{X}_{\sim i})]}{V(Y)} \tag{4}$$

where $V_{\boldsymbol{X}_{\sim i}}[E_{X_i}(Y|\boldsymbol{X}_{\sim i})] = V_{\boldsymbol{X}_{\sim i}}[E_{X_i}(Y|X_1, X_2, \ldots, X_{i-1}, X_{i+1}, \ldots, X_k)] \leq V(Y)$ is the total variance of all input parameters except $X_i$. As the first order index, this total effect index is a value between zero and unity where a value of zero indicates no influence of $X_i$ on the output $Y$ while unity indicates a strong influence (Saltelli et al., 2008). A comprehensive description of the method is given by Saltelli et al. (2008).



For the analysis presented here, the python library SALib (Herman and Usher, 2017) is used for the quasi-random sampling
with low discrepancy after Joe and Kuo (2008) of the input parameter space as well as for the calculation of the Sobol' indices.
It furthermore allows the estimation of the $95\%$ confidence intervals (Herman and Usher, 2021).

### 3.2.2   $\delta$-method

In comparison with the Sobol' indices, the $\delta$-method takes the complete density distribution of the model output into account,
which ensures the conservation of the whole information of the output density distribution (Borgonovo, 2007). The probability
density function of $X_i$ is denoted $f_{X_i}(x_i)$. The sensitivity coefficient $\delta_i$ for the input parameter $X_i$ is calculated using the
marginal density distribution of the input parameter $f_{X_i}(x_i)$ and the difference between the unconditional and the conditional
density functions $f_Y(y)$ and $f_{Y|X_i}(y)$ of the model output with fixed representation $X_i = x_i$ as

$$\delta_i = \frac{1}{2} \int f_{X_i}(x_i) \left[ \int |f_Y(y) - f_{Y|X_i}(y)|\, \mathrm{d}y \right] \, \mathrm{d}x_i \tag{5}$$

(Borgonovo, 2007). $\delta_i$ represents the total effect of an input parameter $X_i$ on $Y$. It can take a value between zero and unity
($0 \leq \delta_i \leq 1$) where zero means that the output is independent of $X_i$ (Plischke et al., 2013). The same library SALib (Herman
and Usher, 2017) was used to apply the $\delta$-method including the estimation of the $95\%$ confidence interval.

### 3.3   Model setup for sensitivity analyses

For the SA a simple model setup with a single source and constant wind was chosen. Figure 1 illustrates the simulation domain
with the plume sampled at a height of $20.5\,\mathrm{m}$ above ground level (agl). The simulation domain has an extent of $10\,\mathrm{km} \times$

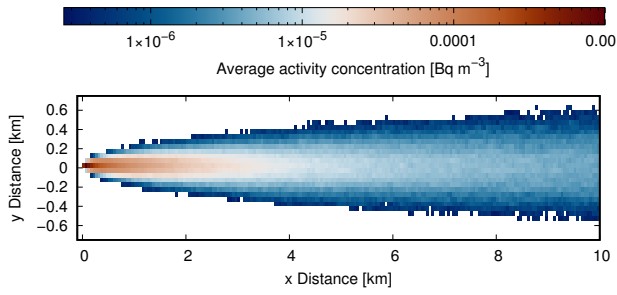

**Figure 1.** X-y-plane of the simulation domain for the sensitivity analyses with activity concentration at a height of $20.5\,\mathrm{m}$. The generic plume
is simulated for SC = neutral, $z_0 = 0.5\,\mathrm{m}$, $d = 6$ and $h_s = 20\,\mathrm{m}$ with a wind speed of $1\,\mathrm{m\,s^{-1}}$ from the west at $10\,\mathrm{m}$ height and an emission
rate of $1\,\mathrm{Bq\,s^{-1}}$. The activity concentration distribution in $\mathrm{Bq\,m^{-3}}$ is given in a logarithmic scale.

$1.5\,\mathrm{km} \times 1.5\,\mathrm{km}$ in x-, y-, and z-direction. The x-direction is defined in west-east orientation, the y-direction in north-south
orientation. The domain is divided into grid cells with a horizontal resolution of $50\,\mathrm{m}$. Vertically, the domain is divided into
19 levels of varying thickness gradually increasing from the lowest layer ($3\,\mathrm{m}$ thick) to the top simulation layer ($300\,\mathrm{m}$ thick).
A table of the level thicknesses is shown in the Table S1. The point source is located at the coordinates ($x = 25\,\mathrm{m}$, $y = 25\,\mathrm{m}$).



**Table 1.** Input parameters and their values and ranges. The default parameters for local SAs are marked with ($^*$).

| Parameter | Values/Range |
|---|---|
| Stability class (SC) [1] | very stable, stable, neutral$^*$, indifferent, unstable, very unstable |
| Roughness length ($z_0$) [1] | $0.10\,\mathrm{m}, 0.20\,\mathrm{m}, 0.50\,\mathrm{m}^*, 1.00\,\mathrm{m}, 1.50\,\mathrm{m}, 2.00\,\mathrm{m}$ |
| Zero-plane displacement factor ($d$) [2] | $3 \ldots 6^* \ldots 15$ |
| Source height ($h_\mathrm{s}$) [3] | $10\,\mathrm{m} \ldots 20\,\mathrm{m}^* \ldots 120\,\mathrm{m}$ |

[1] For the global SAs the given values are sampled.

[2] For the global SAs the values are sampled continuously within the range.

[3] For the global SAs the values are sampled with a resolution of $1\,\mathrm{m}$ within the range.

The vertical position is varied during the SAs. A constant westerly wind ($270°$) was used for the entire simulation period of
24 hours with a velocity of $1\,\mathrm{m\,s^{-1}}$ at $10\,\mathrm{m}$ height. In order to focus on the evolving dispersion pattern, the topography is
assumed as flat surface. The gaseous krypton isotope $^{85}$Kr with a decay constant of $\lambda_\mathrm{decay} = 2.05 \times 10^{-9}\,\mathrm{s^{-1}}$ was used as
tracer. This results in a decay of less than $0.02\,\%$ within the simulation period and can therefore be neglected. The emission
source is represented as a source with a constant activity rate of $1\,\mathrm{Bq\,s^{-1}}$.

The input parameters stability class, roughness length, zero-plane displacement factor and source height are assumed to be
the key parameters of ARTM. These parameters and their ranges are summarised in Table 1. Only a discrete set of six SCs is
allowed in ARTM. The range of $z_0$ values allowed by ARTM is limited to the roughness lengths that correspond the typical
land covers in the vicinity of German nuclear facilities. German authorities recommend a value of the zero-plane displacement
factor $d = 6$ (TA Luft, 2002; VDI 3783 part 8, 2017). The range for the variation of $d$ is centred on this value and limited to
forest canopy heights typical for mixed forest (Lang et al., 2022). In Table 1, the parameter values for the reference point for
the local SAs are marked with * symbol. For global sensitivity analyses, the whole parameter ranges are sampled.

The target quantities of the SAs are two important characteristics of the plume, i) the plume volume, which is a measure
for the tracer dispersion and is closely linked to the maximum mixing ratio and ii) the distance between the location of the
maximum activity concentration and the source position at ground level ,which is of special interest for radiation exposure
assessment.

## 3.4 Results of the sensitivity analyses

The results of the calculations of the local and global sensitivity coefficients are summarised in Table 2. Concerning the plume
volume, all SA methods compute the highest SA coefficients for the stability class. Although less prominent, this is also
observable for the distance between source and maximum concentration at the ground level except for sensitivity indices $SI_i$.

For $SM_{z_0}^{25}$ no value can be calculated because a variation of $\pm 25\,\%$ from the reference roughness length value does not lead
to a change of the categorial $z_0$ value. For the other parameters, the two different ranges of variation ($\alpha = 25$ and $\alpha = 50$)
for $SM_i^\alpha$ provide valuable additional information. For example, it can be seen from Table 2 that the deviations between the





**Table 2.** Sensitivity coefficients of local and global sensitivity analyses for the plume volume and for the distance between the source and the maximum concentration at the ground level. For the Sobol' indices ($S_i$ and $S_{\mathrm{T}i}$) and the $\delta$-method ($\delta_i$) 95 % confidence intervals are given as well. Coefficients with very large relative confidence intervals are marked with ($^*$), coefficients of one method, which cannot be distinguished within their confidence intervals are marked with ($^\dagger$).

| Parameter | $SI_i$ | $SM_i^{25}$ | $SM_i^{50}$ | $S_i$ | $S_{\mathrm{T}i}$ | $\delta_i$ |
|---|---|---|---|---|---|---|
| | | | Plume volume | | | |
| SC | 0.987 | 1 | 1 | $0.981 \pm 0.032$ | $0.99 \pm 0.04$ | $0.666 \pm 0.009$ |
| $z_0$ | 0.718 | - | 0.202 | $0.005 \pm 0.006^*$ | $0.017 \pm 0.002$ | $0.130 \pm 0.001$ |
| $d$ | 0.291 | 0.022 | 0.027 | $(0.3 \pm 8) \cdot 10^{-4*}$ | $(28 \pm 5) \cdot 10^{-5}$ | $0.126 \pm 0.002^\dagger$ |
| $h_\mathrm{s}$ | 0.119 | 0.004 | 0.004 | $(0.9 \pm 9) \cdot 10^{-4*}$ | $(39.5 \pm 2.3) \cdot 10^{-5}$ | $0.126 \pm 0.002^\dagger$ |
| | | | Distance between source and maximum concentration | | | |
| SC | 0.884 | 1 | 1 | $0.16 \pm 0.06$ | $0.90 \pm 0.06$ | $0.328 \pm 0.005$ |
| $z_0$ | 0.769 | - | 0.158 | $0.00 \pm 0.04^*$ | $0.75 \pm 0.07^\dagger$ | $0.118 \pm 0.003^\dagger$ |
| $d$ | 0.222 | 0.250 | 0.053 | $0.00 \pm 0.01^*$ | $0.06 \pm 0.02$ | $0.115 \pm 0.003^\dagger$ |
| $h_\mathrm{s}$ | 0.971 | 0.750 | 0.211 | $0.02 \pm 0.04^*$ | $0.74 \pm 0.07^\dagger$ | $0.129 \pm 0.004$ |

coefficients of $SM_i^{25}$ and $SM_i^{50}$ are small for the plume volume while they are large for the distance between the source and the maximum concentration. The influence of the input parameters seems to be rather linear for the plume volume but highly non-linear for the distance between source and maximum concentration at the ground.

For the global SA methods, both target quantities show a distinct importance not only of the first order (direct influence of one single input parameter) but also of higher order (including interactions of two or more input parameters) effects. A small difference between $S_i$ and $S_{\mathrm{T}i}$ shows a large first order effect as it can be seen for the plume volume. In contrary to this, a large difference reveals a small first order effect compared to a higher order effect as it can be seen for the distance between source and maximum concentration at ground level. This agrees with the conclusions that can be drawn from the $\delta_i$ coefficients. The sum of the sensitivity coefficients for the plume volume $\sum_i \delta_i = 1.05 \pm 0.01 \cong 1$ indicates that the effects of variation in the input parameters on variation in the plume volume are separable, i.e. interactions between input parameters play a minor role. For the distance between source and maximum concentration, the sum of the sensitivity coefficients $\sum_i \delta_i = 0.690 \pm 0.008 \ncong 1$ indicates the important role of cross interactions between the input parameters (Borgonovo, 2007). In contrary to the findings of the Sobol' indices that some input parameters having negligible influences, the $\delta$-method suggests that the output characteristics are sensitive to all parameters. This difference could be due to the different amounts of information processed by the two methods. While the Sobol' indices compare conditional and unconditional variances of the output distribution the $\delta$-method takes the entire output distribution into account.




Some of the global SA coefficients have very large relative confidence intervals and cannot be distinguished from zero (marked with $^*$). Others cannot be distinguished from each other within their confidence intervals (marked with $^\dagger$). Increasing the sample size of 24,576 further would be necessary to get smaller confidence intervals but this would also increase the computation time (Herman and Usher, 2021).

Based on the coefficients from Table 2, the input parameters were ranked according to their importance as summarised in Table 3. The rankings obtained for the individual SA methods differ not only for the two target quantities but also between different methods. The overall ranking, which is simply computed as the sum ($\Sigma$) over the different methods, is provided in the last column.

**Table 3.** Ranking of the influence of the input parameters on the plume volume and on the distance between the source and the maximum concentration at ground level for local and global sensitivity analysis methods.

| Parameter | $SI_i$ | $SM_i^{25}$ | $SM_i^{50}$ | $S_i$ | $S_{\mathrm{T}i}$ | $\delta_i$ | $\Sigma$ | Rank |
|---|---|---|---|---|---|---|---|---|
| | | | Plume volume | | | | | |
| SC | 1 | 1 | 1 | 1 | 1 | 1 | 6 | 1 |
| $z_0$ | 2 | 3 | 2 | 3 | 2 | 2 | 14 | 2 |
| $d$ | 3 | 2 | 3 | 3 | 4 | 3.5 | 18.5 | 3 |
| $h_{\mathrm{s}}$ | 4 | 4 | 4 | 3 | 3 | 3.5 | 21.5 | 4 |
| | | Distance between source and maximum concentration | | | | | | |
| Parameter | $SI_i$ | $SM_i^{25}$ | $SM_i^{50}$ | $S_i$ | $S_{\mathrm{T}i}$ | $\delta_i$ | $\Sigma$ | Rank |
| SC | 2 | 1 | 1 | 1 | 1 | 1 | 7 | 1 |
| $z_0$ | 3 | 3 | 3 | 3 | 2.5 | 3.5 | 18 | 3 |
| $d$ | 4 | 4 | 4 | 3 | 4 | 3.5 | 22.5 | 4 |
| $h_{\mathrm{s}}$ | 1 | 2 | 2 | 3 | 2.5 | 2 | 12.5 | 2 |

The most unambiguous result is that all SA methods show the plume volume to be most sensitive to the SC. The ranks for the other input parameters, in contrast, are not the same. At this point we want to mention that the ranking of $SM_i^{25}$ given in Table 3 is the average of all possible rankings for this method when taking into account that there is no coefficient for $SM_{z_0}^{25}$. The rankings of the remaining local SA methods $SI_i$ and $SM_i^{50}$ are in agreement with each other for the plume volume, while the rankings of the global SA methods disagree. Compared to the rankings for the plume volume, those for the distance between source and maximum concentration at the ground level are less uniform.

The overall rankings for both target quantities differ from each other, which emphasises that different target quantities are not necessarily sensitive to the same input parameters. Both target quantities are most sensitive to the SC, which is thus a potential source of high uncertainty. The source height $h_{\mathrm{s}}$ has little influence on the plume volume but it is the second most



important parameter for the distance between source and maximum concentration at the ground level. The strong influence of $h_s$ on this target quantity is intuitively understandable, but interestingly SC is still more important.

## 4 Turbulence models and their performance on the well-mixed test

In LPDMs the turbulent motion is described via a Markov chain approach in the form of a Langevin equation (Lin and Gerbig, 2013). ARTM uses the wind speed fluctuation $\sigma$ and Lagrangian correlation time $T_L$ as input parameters for this Markov chain approach (Hanfland et al., 2022). The turbulence model specifies these two parameters and thus, influences the tracer dispersion and hence the simulated mixing ratio field (Katharopoulos et al., 2022).

### 4.1 Description of turbulence models

The turbulence model implemented in ARTM 2.8.0 as the default model is not widely used in the scientific community. Besides this, it has been reported by Janicke and Janicke (2011) that it sometimes underestimates plume dispersion. Therefore, they introduced a modified turbulence model leading to stronger dispersion, which can optionally be activated in ARTM. In 2022, the new version 3.0.0 of ARTM was released. It implements a new turbulence model according to the Association of German Engineers (VDI) guideline VDI 3783 part 8 (2017). All three models deviate from the model suggested by Hanna (1982), which is quite widely used and thoroughly tested against tracer release experiments. However, in this model the turbulence may abruptly change between SCs. To overcome this issue of discontinuity, Degrazia et al. (2000) proposed a continuous description of the turbulence throughout all atmospheric conditions. The wind speed fluctuation $\sigma$ and the Lagrangian correlation time scales $T_L$ of the five turbulence models are presented in the following Eqs. 6 - 26 for unstable stratification and their profiles are displayed in Fig. 2. For the following quantities we define the x-components along the average horizontal wind direction, the y-components perpendicular to it in the horizontal plane and the z-components in the vertical direction. For simplicity the zero-plane displacement is not taken into account in the following equations.

The first model, the default boundary layer model (BLM) of ARTM 2.8.0, describes profiles for the wind speed fluctuations as

$$\sigma_x = 2.4 \cdot u_* \left( 1 + 0.01486 \frac{-h_m}{\kappa L} \right)^{\frac{1}{3}} \cdot \exp\left( \frac{-z}{h_m} \right), \tag{6}$$

$$\sigma_y = 1.8 \cdot u_* \left( 1 + 0.03522 \frac{-h_m}{\kappa L} \right)^{\frac{1}{3}} \cdot \exp\left( \frac{-z}{h_m} \right) \tag{7}$$

and

$$\sigma_z = 1.3 \cdot u_* \left[ \left( 1 - 0.8 \frac{z}{h_m} \right)^3 \cdot \frac{-z}{\kappa L} + \exp\left( \frac{-z}{h_m} \right)^3 \right]^{\frac{1}{3}} \tag{8}$$





**Figure 2.** Vertical profiles of model characteristics for the five turbulence models ARTM2, ARTM3, PRFMOD, MODHANNA and DE-GRAZIA for very unstable atmospheric conditions. Wind speed fluctuations $\sigma$: a) along wind direction $\sigma_x$; c) in the crosswind direction $\sigma_y$; and e) in the vertical direction $\sigma_z$ against the normalized height (normalized to the boundary layer height). The corresponding Lagrangian correlation times $T_L$ are shown in b), d) and f), respectively. The turbulent kinetic energy per unit mass $TKE$ is shown in g).



where $u_*$ is the friction velocity, $h_\mathrm{m}$ is the mixing layer height, $\kappa = 0.4$ is the von Kármán constant, $L$ is the Obukhov length and $z$ is the height agl (Kerschgens et al., 2000; VDI 3783 part 8, 2002; Hanfland et al., 2022). This model is called ARTM2 in the following.

250  The second turbulence model available in ARTM is based on ARTM2 with a modification in the exponents as well as in the prefactor of the crosswind component as

$$\sigma_\mathrm{x} = 2.4 \cdot u_* \left( 1 + 0.01486 \frac{-h_\mathrm{m}}{\kappa L} \right)^{\frac{1}{3}} \cdot \exp\left( \frac{-0.3 \cdot z}{h_\mathrm{m}} \right), \tag{9}$$

$$\sigma_\mathrm{y} = 2.0 \cdot u_* \left( 1 + 0.03522 \frac{-h_\mathrm{m}}{\kappa L} \right)^{\frac{1}{3}} \cdot \exp\left( \frac{-0.3 \cdot z}{h_\mathrm{m}} \right) \tag{10}$$

255  and

$$\sigma_\mathrm{z} = 1.3 \cdot u_* \left[ \left( 1 - 0.8 \frac{z}{h_\mathrm{m}} \right)^3 \cdot \frac{-z}{\kappa L} + \exp\left( \frac{-0.3 \cdot z}{h_\mathrm{m}} \right)^3 \right]^{\frac{1}{3}} \tag{11}$$

(Janicke and Janicke, 2011). This model leads to wider plumes and is called PRFMOD in the following.

In addition to the two previous models we added a turbulence model to ARTM based on the formulations used in other ADMs (Stohl et al., 2005). This model uses $\sigma_z$ from ARTM2 given in Eq. 8 but the horizontal wind speed fluctuations

260 $$\sigma_\mathrm{x} = \sigma_\mathrm{y} = u_* \left( 12 + \frac{h_\mathrm{m}}{2|L|} \right)^{\frac{1}{3}} \tag{12}$$

are equal to the equations suggested by Hanna (1982). In the following, this model is called MODHANNA.

The Lagrangian correlation times of the three models above are given according to Kolmogorov's theory as

$$T_{\mathrm{L}i} = \frac{2 \cdot \sigma_i^2}{\mathrm{C}_0 \cdot \eta} \tag{13}$$

(Luhar and Britter, 1989) with the Kolmogorov constant $\mathrm{C}_0 = 5.7$ and the dissipation rate of the turbulent kinetic energy

265 $$\eta = \max\left\{ \frac{u_*^3}{\kappa z} \left[ \left( 1 - \frac{z}{h_\mathrm{m}} \right)^2 + \frac{z}{h_\mathrm{m}} \right] + \frac{-u_*^3}{\kappa L} \left[ 1.5 - 1.3 \left( \frac{z}{h_\mathrm{m}} \right)^{\frac{1}{3}} \right], \frac{u_*^3}{\kappa z} \right\}. \tag{14}$$

The fourth model is the default model of the new version 3.0.0 of ARTM with the wind speed fluctuations given as

$$\sigma_\mathrm{x} = 2.4 \cdot u_* \left[ 1 + 0.01486 \frac{-h_\mathrm{m}}{\kappa L} \cdot \exp\left( -0.9 \frac{z}{h_\mathrm{m}} \right) \right]^{\frac{1}{3}}, \tag{15}$$

$$\sigma_\mathrm{x} = 2.0 \cdot u_* \left[ 1 + 0.02568 \frac{-h_\mathrm{m}}{\kappa L} \cdot \exp\left( -0.9 \frac{z}{h_\mathrm{m}} \right) \right]^{\frac{1}{3}} \tag{16}$$

270 and

$$\sigma_\mathrm{z} = 1.3 \cdot u_* \left[ \left( 1 - 0.8 \frac{z}{h_\mathrm{m}} \right)^3 \cdot \frac{-z}{\kappa L} + \exp\left( -0.9 \frac{z}{h_\mathrm{m}} \right)^3 \right]^{\frac{1}{3}} \tag{17}$$



(VDI 3783 part 8, 2017). The Lagrangian correlation time scales are calculated via the turbulent diffusion coefficients $K_i$ as

$$T_{\mathrm{L}i} = \frac{K_i}{\sigma_i^2} \tag{18}$$

with

$$K_j = 0.9 \frac{u(z) \cdot h_{\mathrm{m}}}{100 \cdot u_*} \sigma_j \tag{19}$$

for the horizontal components $j$ and

$$K_z = \kappa u_* z \left[ \left( 1 - 0.8 \frac{z}{h_{\mathrm{m}}} \right)^4 \frac{9z}{-L} + \exp\left( -3.6 \frac{z}{h_{\mathrm{m}}} \right) \right]^{\frac{1}{2}} \tag{20}$$

for the vertical component, respectively (VDI 3783 part 8, 2017). This model is called ARTM3 in the following.

We implemented a fifth model, which in contrast to the previous four turbulence models that are based on similarity theory, is based on the spectral distribution of the turbulent kinetic energy of the boundary layer and was presented by Degrazia et al. (2000). For very unstable boundary conditions the wind speed fluctuations are given as

$$\sigma_{\mathrm{x}} = 0.53 \cdot u_* \left( \frac{-h_{\mathrm{m}}}{\kappa L} \right)^{\frac{1}{3}}, \tag{21}$$

$$\sigma_{\mathrm{y}} = 0.61 \cdot u_* \left( \frac{-h_{\mathrm{m}}}{\kappa L} \right)^{\frac{1}{3}} \tag{22}$$

and

$$\sigma_{\mathrm{z}} = 0.54 \cdot u_* \left( \frac{-h_{\mathrm{m}}}{\kappa L} \right)^{\frac{1}{3}} \cdot \left\{ 1.8 \left[ 1 - \exp\left( \frac{-4z}{h_{\mathrm{m}}} \right) - 0.0003 \cdot \exp\left( \frac{8z}{h_{\mathrm{m}}} \right) \right] \right\}^{\frac{1}{3}} \tag{23}$$

with the Lagrangian correlation times

$$T_{\mathrm{L}i} = \frac{l_i}{\sigma_i} \tag{24}$$

where $l_i$ is the Lagrangian correlation length given as

$$l_{\mathrm{x}} = l_{\mathrm{y}} = 0.21 \cdot h_{\mathrm{m}} \left( 0.01 \frac{h_{\mathrm{m}}}{-L} \right)^{\frac{1}{2}} \tag{25}$$

and

$$l_{\mathrm{z}} = 0.14 \cdot h_{\mathrm{m}} \left( 0.01 \frac{h_{\mathrm{m}}}{-L} \right)^{\frac{1}{2}} \cdot \left\{ 1.8 \left[ 1 - \exp\left( \frac{-4z}{h_{\mathrm{m}}} \right) - 0.0003 \cdot \exp\left( \frac{8z}{h_{\mathrm{m}}} \right) \right] \right\}. \tag{26}$$

In this work this turbulence model is denoted as DEGRAZIA.

The turbulent kinetic energy per unit mass is determined as

$$TKE = \frac{1}{2} \left( \sigma_{\mathrm{x}}^2 + \sigma_{\mathrm{y}}^2 + \sigma_{\mathrm{z}}^2 \right) \tag{27}$$

(Stull, 1988).





## 4.2 Evaluation of turbulent mixing

The degree of preservation of well-mixed conditions is a key quality indicator for any LPDM, similar to the preservation of mass in a Eulerian model. It tests whether an initially uniform distribution of a tracer in an incompressible flow remains uniform as postulated by the second law of thermodynamics (Sawford, 1986; Thomson, 1987; Lin and Gerbig, 2013; Bahlali et al., 2020). Exactly fulfilling this criterion is challenging, but it is important to quantify the degree of deviation from this ideal behaviour to judge the magnitude of systematic model biases and whether these biases are acceptable. In this work the tests of the turbulence models are limited to the case of very unstable atmospheric conditions because the observations that are used for the comparison were collected at very unstable atmospheric conditions, too.

The well-mixed condition test can characterise the vertical mixing homogeneity of a model. For these tests simulation domains with periodic horizontal boundaries and reflecting vertical boundaries are used This virtually expands the simulation domain to infinite extent and prevents the simulation from losing tracer mass. The whole simulation domain serves as a volume source where 115,200 numerical particles are inserted uniformly within the first simulation hour because ARTM does not take into account the changing density with height. The domain size is $2000\,\mathrm{m} \times 2000\,\mathrm{m} \times 1100\,\mathrm{m}$ in x-, y-, and z-direction with a horizontal (vertical) resolution of $200\,\mathrm{m}$ ($25\,\mathrm{m}$), respectively. The vertical extent of the domain is equal to the assumed mixing depth. A temporally constant wind profile with a wind speed of $1\,\mathrm{m\,s^{-1}}$ at $10\,\mathrm{m}$ height and a direction of $270°$ (westerly) is used. For the evaluation, the hourly mean concentration and its standard deviation was derived for each vertical level.

The concentration profiles of the different turbulence models for very unstable PBL conditions are shown in Fig. 3. The concentrations of the state of mixing after one hour (red line) and after two hours (blue dashed line) are shown. Concentration values are normalised to the mean concentration ($c\,\bar{c}^{-1}$) and the height is normalised to the mixing depth ($z\,h_{\mathrm{m}}^{-1}$). We used the same initial numerical particle distribution for all turbulence models to eliminate possible differences arising from different initial distributions.

The concentration profiles after one hour differ from the uniform distribution $c\,\bar{c}^{-1} = 1$. This indicates a certain degree of segregation of the numerical particles but most deviations are less than $5\%$ (vertical dashed lines). The largest deviations can be found at the top of the PBL for the ARTM3 model ($> 30\%$) and the DEGRAZIA model ($> 15\%$). The profiles of the ARTM2 and the MODHANNA turbulence models are very similar since they both contain the same vertical turbulence parametrisation. The PRFMOD turbulence model differs slightly from the ARTM2 model due to modifications described in Eq. 11. The profile of ARTM3 shows trends of dilution and accumulation similar to ARTM2, PRFMOD and MODHANNA but magnified in its extent. The profile of the DEGRAZIA turbulence model shows a different shape because of its different formulation of the turbulence parameters (see Eqs. 23, 24 and 26).

By $t = 2\,\mathrm{h}$, the dilution of concentration has further increased at the bottom and the top of the PBL and the accumulation at $z\,h_{\mathrm{m}}^{-1} \approx 0.3$ (horizontal dashed black line) has further increased partly beyond $5\%$ but well below $10\%$ deviation for ARTM2, PRFMOD, MODHANNA and DEGRAZIA. For ARTM3, the dilution at the ground almost vanishes while the dilution above $z\,h_{\mathrm{m}}^{-1} = 0.8$ increases to $40\%$ and the accumulation in the middle of the PBL increases to $18\%$. After the second hour, no





**Figure 3.** Profiles of the concentration normalised to the mean concentration $c\overline{c}^{-1}$ (a, b, c, d, e) of the different turbulence models ARTM2, ARTM3, PRFMOD, MODHANNA and DEGRAZIA after one hour (red lines) and two hours (blue dashed dotted lines) for periodic lateral simulation domain boundaries and reflecting bottom and top boundaries. In b) the x-axis scale changes at $c\overline{c}^{-1} = 0.9$. f) Time series of the normalised concentration at normalised height $z\,h_{\mathrm{m}}^{-1} \approx 0.3$ for the ARTM2 model, which is indicated by the dashed horizontal line in a). The x-axis scale changes at 10 hours.

330    further changes are observed as it can be seen in Fig. 3 f) for the ARTM2 turbulence model at $z\,h_{\mathrm{m}}^{-1} \approx 0.3$. Time series for other heights and other turbulence models show similar behaviour and are given in Supplement S2.

This well-mixed condition test shows that the simulation result systematically overestimates the concentration values at $z\,h_{\mathrm{m}}^{-1} \approx 0.3$ for the ARTM2, PRFMOD and the MODHANNA model after the second hour. Near the surface, which is im-





portant for estimation of exposure to the population, the concentration values are underestimated. In both cases, the errors are
only $5 - 6\%$. At the top of the PBL, the models underestimate the expected concentration significantly. The ARTM3 turbulence
model shows the smallest deviation from the mean domain concentration near the ground but it overestimates the concentration
in the middle of the PBL before turning into a substantial underestimation towards the mixing layer top. Below $z\,h_{\mathrm{m}}^{-1} = 0.8$
the turbulence model DEGRAZIA performs best. At the top of the PBL the model decreases well below the expected concen-
tration. All the tested turbulence models can be assumed as acceptable for simulations at very unstable atmospheric conditions
but the partly large deviations of the concentration from the expected values at certain heights have to be taken into account
when interpreting model results.

## 5 Comparison of ARTM simulation with airborne observations

The comparison of atmospheric dispersion simulation results with measurements near the ground is not sufficient to derive
any conclusions about the three-dimensional structure of simulated emission plumes. To study the agreement of simulated
and observed plume dispersion it is inevitable to use observations that resolve the structure of the real plume. Since ARTM
simulates the emissions of nuclear facilities from quite high stack, it is useful to choose observational data originating from
similar height levels. In this work, we present a comparison of ARTM simulations with airborne $CO_2$ observations within the
PBL. In such a case, the comparison is challenging because of the turbulent character of the PBL. As pointed out by Brunner
et al. (2022), observations only provide snap-shots of the real world while simulations provide one realisation of a multitude
of stochastic representations of the real world. Simulations with slightly perturbed initial conditions could result in different
dispersion patterns of the plume. Furthermore, simulation results and observations may have different spatial and temporal
resolutions and uncertainties, which complicate the comparison of simulations with observations (Farchi et al., 2016). Thus,
in this work, the comparison of simulation results with observations for five turbulence models are given using rather general
plume characteristics such as the plume width per transect and maximum mixing ratios.

### 5.1 Observational data

The aircraft observations used for this investigation originate from the Carbon Dioxide and Methane Mission (CoMet 1.0) (Fix
et al., 2018; Luther et al., 2019; Fiehn et al., 2020; Gałkowski et al., 2021; Wolff et al., 2021; Krautwurst et al., 2021; Andersen
et al., 2022; Brunner et al., 2022). The campaign took place in May and June 2018 and involved three aircraft performing in
situ and remote sensing measurements. The objective was to study $CO_2$ and methane ($CH_4$) emissions from different sources
in Europe including power plants, as well as to compare the different observational methods.

For the evaluation of ARTM, airborne in situ $CO_2$ measurements in the vicinity of the Bełchatów lignite power plant in
Poland were used (Fiehn et al., 2020; Kostinek et al., 2021). An overview map with the $CO_2$ mixing ratios along the flight path
is shown in Fig. 4. The in situ measurements had been performed on 7 June 2018 between 13:00 and 15:00 UTC aboard the
DLR Cessna Grand Caravan 208B. One transect on the upwind side of the emitter was performed at the beginning in order to
derive the mean background $CO_2$ mixing ratio $c_{CO_2} = 401.2\,\mathrm{ppmv}$. The exhaust plume of the power plant was probed during





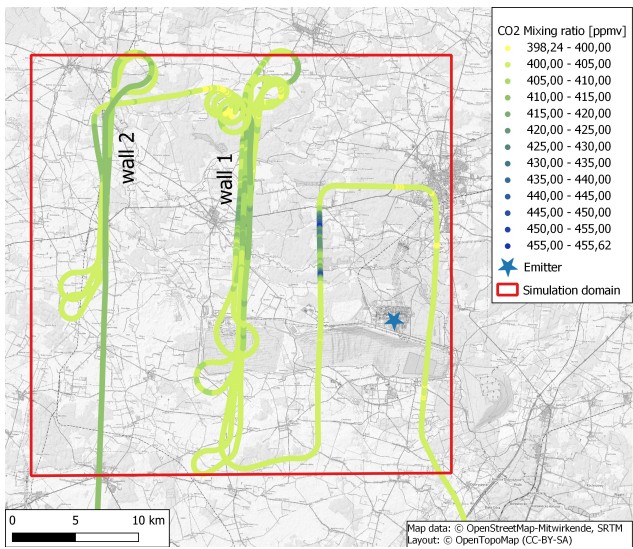

**Figure 4.** Map showing the flight path of the DLR Cessna aircraft in the vicinity of the Bełchatów lignite power plant (blue star), color-coded by the in situ measured $CO_2$ values. Transects were performed both east (upwind side) and west (downwind side) of the emitting power plant. The red box indicates the simulation domain.

several transects on the downwind side at heights between $500\,\text{m}$ and $1.7\,\text{km}$ agl. They form two wall patterns at meridional distances of approx. $13\,\text{km}$ (wall 1) and $23\,\text{km}$ (wall 2) and a single transect at approx. $6\,\text{km}$ from the source.

The $CO_2$ had been measured with a cavity ring-down spectroscopy analyser (G1301-m, Piccaro) specifically modified for the airborne deployment and with an uncertainty of $\pm 0.15\,\text{ppmv}$. Details of the measurement equipment are described by
Klausner et al. (2020). Observational data for wind direction, wind speed and flight height are shown in Supplement S3.

## 5.2 Model setup

We chose a simulation domain of $33.3\,\text{km} \times 33.3\,\text{km} \times 1.9\,\text{km}$ that covers the horizontal extent of the flight trajectory and vertically extends beyond the mixing layer depth by four simulation level. The horizontal resolution was $150\,\text{m}$. The extent of the simulation domain with the location of the emission source (two stacks in a distance of $300\,\text{m}$) is shown in Fig. 4. Vertically,
the grid spacing gradually increases from $3\,\text{m}$ to $35\,\text{m}$ until $100\,\text{m}$ height is reached. Above, $50\,\text{m}$ level thickness was used. All level thicknesses are given in Table S2.

ARTM requires several input parameters: SC, $z_0$, $d_0$, orography, several source specific parameters as well as wind speed and direction at one location in the simulation domain. Since there were no stationary ground based wind measurements available, wind direction and wind velocity as well as SC and mixing layer height were derived from the airborne measurements. The
actual emission rates are unknown. However, Brunner et al. (2022) estimated the overall $CO_2$ emission rate according to the generated electrical power of the power plant resulting in $1503.0\,\text{kg}\,\text{s}^{-1}$ during the measurement flight. This corresponds to $123\%$ of the annual mean emission rate of $38.4\,\text{Mt}\ CO_2$ reported by the power plant to the European Pollutant Release and



**Table 4.** Input parameters needed by ARTM that are constant during the simulation run.

| Parameter | Value | Reference |
|---|---|---|
| Stability class | very unstable | (KTA 1508, 2017) |
| Roughness length | $0.50\,\mathrm{m}$ | (TA Luft, 2002) |
| Zero plane displacement | $6 \cdot 0.50\,\mathrm{m}$ | (TA Luft, 2002) |
| Mixing layer height | $1650\,\mathrm{m}$ | - |
| Stack heights | $300\,\mathrm{m}$ | (Emporis, 2000) |
| Plume rise (western stack) | $202\,\mathrm{m}$ | - |
| Plume rise (eastern stack) | $74\,\mathrm{m}$ | - |
| Emission rate (western source) | $1002.0\,\mathrm{kg\,s^{-1}}$ | (Brunner et al., 2022) |
| Emission rate (eastern source) | $501.0\,\mathrm{kg\,s^{-1}}$ | (Brunner et al., 2022) |
| Orography | SRTM3 data | (Farr et al., 2007) |

Transfer Register (E-PRTR) for the year 2018. The description of the derivation of SC, $z_0$, $d_0$, and the plume rise as well as the orography data are given in Supplement S4. The parameter values and the origin of the orography data are summarised in
Table 4.

ARTM requires radionuclide emission rates in $\mathrm{Bq\,s^{-1}}$. As tracer $^{14}\mathrm{C}$ in its bounded form as $CO_2$ is used. Its decay constant $\lambda = 5730 \pm 40$ years leads to a decay of $5.5 \times 10^{-6}\,\%$ which is negligible within the simulation period. Thus, ARTM's internal emission rates in $\mathrm{Bq\,s^{-1}}$ can be used as an equivalent for a mass rate in $\mathrm{kg\,s^{-1}}$ and to convert activity concentration into mixing ratio.

The wind speed ($4.4\,\mathrm{m\,s^{-1}}$) and directions driving the simulation were derived from one flight transect (13:28:03 UTC to 13:33:14 UTC) at a distance of $\approx 13\,\mathrm{km}$ to the west of the power plant at a height of $\approx 600\,\mathrm{m}$ agl. This transect is located close to the middle of the simulation domain and therefore assumed to be representative. The histogram of the wind directions of the transect is shown in Fig. 5. Based on this histogram, two different setups of the model were selected:

i) A single wind direction of $120°$ (mean of the distribution) was selected, assuming that the wind fluctuations are part of
the turbulence spectrum and should therefore be represented by the turbulence parametrisation of ARTM.

ii) Two different wind directions were used alternatingly to drive ARTM, a direction of $106°$ (mean of all directions $< 120°$) and a direction of $134°$ (mean of all directions $> 120°$). This assumes that part of the wind variation is due to meso-scale variability that cannot be resolved by ARTM's turbulence scheme.

The first setup was applied for all turbulence models while the second setup was only tested for ARTM2. The hourly
sequence of wind inputs for the model is summarised in Supplement S5.



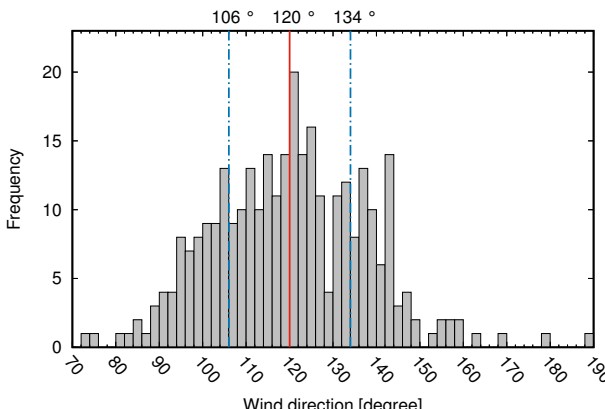

**Figure 5.** Histogram of the wind directions of the transect chosen for the determination of the wind direction and wind velocity. The transect covers a duration from 13:28:03 UTC to 13:33:14 UTC with the mean position $53.31°$ N, $19.15°$ E. The mean measurement height is $599\,\text{m}$ agl. The mean value of the wind direction is $120°$ (red line). The mean value for the wind directions below $120°$ is $106°$, above $120°$ is $134°$ (blue dashed lines), respectively.

### 5.3 Horizontal dispersion

The mixing ratio maps simulated with the five turbulence models at a height of $750\,\text{m}$ to $800\,\text{m}$ are shown in Fig. 6 together with the observations between $700\,\text{m}$ to $800\,\text{m}$. We subtracted the background $CO_2$ mixing ratio of $401\,\text{ppmv}$ from the observation to make them comparable to the simulation results.

The simulated and observed mixing ratios of the plumes are in the same order of magnitude. The simulated plumes show the mean wind direction to be in agreement with the observed one, however, the meandering behaviour of the real plume can be observed at transect 1, 2 and 3 in Fig. 6 revealing that this behaviour is not covered by all turbulence models. The mixing ratio profile in lateral (crosswind) direction simulated by ARTM resembles a Gaussian distribution. This is expected for a constant wind direction and wind speed (Thykier-Nielsen et al., 1999).

The different turbulence models clearly affect the simulated plume widths. The ARTM2 turbulence model simulates the narrowest plume. The ARTM3 model results in a slightly wider plume but compared to the observations both are too narrow. The PRFMOD and DEGRAZIA turbulence models show much broader plumes that cover the observed one to a large extent. The widest plume is simulated by the MODHANNA turbulence model and is in good agreement with the observed plume width. The width of the plumes of the turbulence models is mainly attributed to the horizontal wind speed fluctuations and

Lagrangian correlation times displayed in Fig. 2. The highest values for $\sigma_y$ and $T_{\text{L}y}$ are simulated by MODHANNA, PRFMOD and DEGRAZIA followed by ARTM3 and ARTM2 in the upper half of the PBL in agreement with the simulated plume width.

Fig. 7 shows the simulated and observed plumes of the different turbulence models together with the flight height agl along the flight path. Transects 1, 2 and 3 are shaded grey. Data above the simulated boundary layer top is excluded from the figures. In Fig. 7, the simulated maximum $CO_2$ mixing ratios of ARTM2 are at all transects larger compared to the observations.





**Figure 6.** Modelled $CO_2$ mixing ratio for the case of one wind direction in a) ARTM2, b) ARTM3, c) PRFMOD, d) MODHANNA and e) DEGRAZIA; and two wind directions in f) ARTM2. The wind directions and speeds are given in Table S3; the input parameters in Table 4. Mixing ratios of the simulated plume (averaged over the simulation time) at heights between $750\,\mathrm{m}$ to $800\,\mathrm{m}$ and the in situ data along the flight path between $700\,\mathrm{m}$ to $800\,\mathrm{m}$ is shown in logarithmic scale in ppmv. The two wind direction case in f) shows the mean $CO_2$ mixing ratio of two subsequent hours for the duration of the measurement flight from 13:00 UTC to 15:00 UTC. The background $CO_2$ mixing ratio of $401\,\mathrm{ppmv}$ is subtracted from the observation.



**Figure 7.** Simulated (red line) and measured (black line) $CO_2$ data along the flight path together with the flight height (blue dotted line) within the simulation domain. a) ARTM2, b) ARTM3, c) PRFMOD, d) MODHANNA, e) DEGRAZIA turbulence models and f) ARTM2 turbulence model with two alternating wind directions. The transects shown in Fig. 6 are shaded grey.



Within the simulated boundary layer this deviation reaches 300% at 14:08 UTC and is attributed to the too narrow simulated plume. With increasing plume width of the different turbulence models the maximum mixing ratios decrease (see Figs. 7 a – e). The turbulence models ARTM3, PRFMOD, MODHANNA and DEGRAZIA simulate mixing ratio peaks similar to or below the observed values. Due to dispersion the mixing ratio maximums decrease with increasing distance from the source for all models in agreement with the observation. It is important to point out that simulated mixing ratio values are highly dependent

on the emission rates.

    The simulation gives one hour averages of the exhaust plume, which is expected as the mean of several realisations of meandering plumes. It is not expected that simulated values are much larger than the observed ones but can occur if the width of the simulated plume or the mixing layer depth are underestimated or the emission rate is overestimated.

    An alternative to model the meandering behaviour via the turbulence is the usage of alternating wind directions for sub-

sequent simulation hours for the ARTM2 turbulence model to explicitly simulate the meandering plume (Figs. 6 f and 7 f). Simulation results from subsequent hours are combined by calculating the average concentrations. The wind direction derivation is explained in Sec. 5.2. This method generates the widest plume covering the observations and mimics the structure of two maxima at transect 1. However, these two observed maxima originate from snap-shots of the meandering plume and is not expected to be reproduced by the time-averaged simulation. Moreover, physically unrealistic plateaus of mixing ratios are

simulated in wall 1 and a single narrow mixing ratio peak in wall 2 which is a result of the alternating wind directions. Mixing ratio maps of simulations and observations at other selected heights are given in Supplement S6.

## 5.4   Vertical dispersion

For the analysis of the vertical plume behaviour, the cross sections of the simulated plumes at wall 1 are presented in Fig. 8. The narrowest simulated plume is obtained by the ARTM2 model and underestimates the width of the observed plume at

heights from $600\,\mathrm{m}$ to $1400\,\mathrm{m}$ agl. The plume of ARTM3 model is slightly wider throughout the PBL. In both, the ARTM3 and PRFMOD model, the values of $\sigma_\mathrm{y}$ ($T_{\mathrm{L}y}$) decrease (increase) with height, respectively (see Fig.2). While these opposing trends cancel out each other for the ARTM3 model they lead to a slightly increase of lateral dispersion with height for the PRFMOD model. The vertical profiles of $\sigma_y$ and $T_{\mathrm{L}y}$ of the MODHANNA model shown in Figs. 2 c) and d) appear to lead to a slightly increasing dispersion, too. The DEGRAZIA model in contrast, shows a constant behaviour for both, $\sigma_\mathrm{y}$ and $T_{\mathrm{L}y}$. Below $200\,\mathrm{m}$

the width of all five simulated plumes decrease towards the surface.

    All turbulence models show a slight decrease of the mixing ratio with increasing height at a constant distance from the source (see Fig. 7), which agrees with observations. From the cross sections at wall 1 (Fig. 8) the average horizontal mixing ratio profiles are derived and shown in Fig 9. Except for the DEGRAZIA model, the decreasing mixing ratio with increasing height above $600\,\mathrm{m}$ can be recognised here as well. However, in Fig. 7 the highest maximum mixing ratios at wall 1 occur at

different transects for the simulations and the observation. This indicates that the effective source height of the simulations was assumed too low compared to the measurements.



**Figure 8.** Cross section of the simulated plumes at wall 1 of the observations for the different turbulence models a) ARTM2, b) ARTM3, c)PRFMOD, d)MODHANNA and e) DEGRAZIA. f) the two wind directions case for ARTM2. The x-axis "y Distance" is in south-north orientation. The dashed line at 1650 m agl marks the simulated mixing layer top.





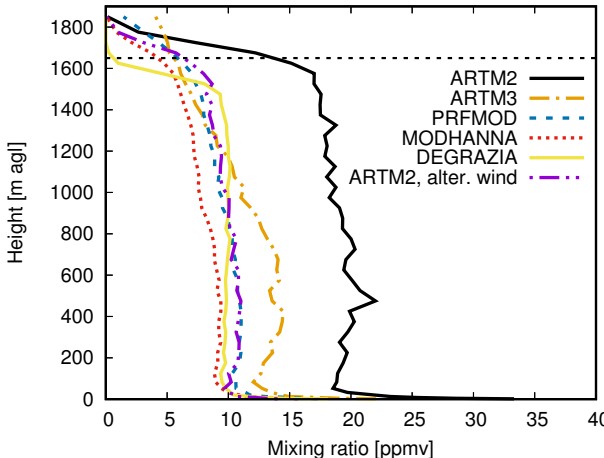

**Figure 9.** Profile of the average horizontal mixing ratio of the six simulation cases at wall 1 (see Fig. 8). The dashed line at $1650\,\mathrm{m}$ agl marks the simulated mixing layer top.

In contrast to the Gaussian lateral mixing ratio distribution of the plume of Fig. 8 a), the ARTM2 turbulence model with two alternating wind directions (Fig. 8 f) shows the uniform mixing ratio distribution in the plume's core region (mixing ratio > $1\,\mathrm{ppmv}$) as it was already shown in Fig. 7 f.

The cross sections of wall 2 in Fig. S11 of the Supplement show a similar behaviour of the plumes. The measured data show a large variation of the plume width on the different transects emphasising the meandering and turbulent character of the real plume. Furthermore, it can be recognised that the real plume is not entirely recorded; the transects are too short at this wall.

**5.5    Validation and uncertainty evaluation**

In order to quantify the simulations uncertainty, we investigate the deviations of the simulated and the observed $CO_2$ mixing
ratios in the plume by probability distributions (PDs), comparisons of integrated plume mass and point-to-point mixing ratio comparisons.

The deviation of model results and measurements in a plume can be accessed by the comparison of the PDs and the cumulative probability distributions (CDPs) of simulated and observed $CO_2$ mixing ratios in the plume. The PDs of simulation and measurement are normalised to the maximum mixing ratio of the measurements with the integrals of simulated and measured
distributions being equal. To get rid of the mixing ratio fluctuation of the excess mixing ratio of the measurement, mixing ratio values below $1\,\mathrm{ppmv}$ are not taken into account. The PDs and CPDs of the five different turbulence models and the observation for all transects below the simulated boundary layer top are given in Fig. 10. The PDs of the simulated and measured plume show the occurrence of mixing ratio values relative to the maximum mixing ratio of the measurements. There is an overestimation of simulated maximum mixing ratios for the ARTM2 and ARTM3 turbulence model. The high number of data points at
approx. 20% of the maximum mixing ratio of the measurements is due to the fine structure, shoulders beside peaks and broad



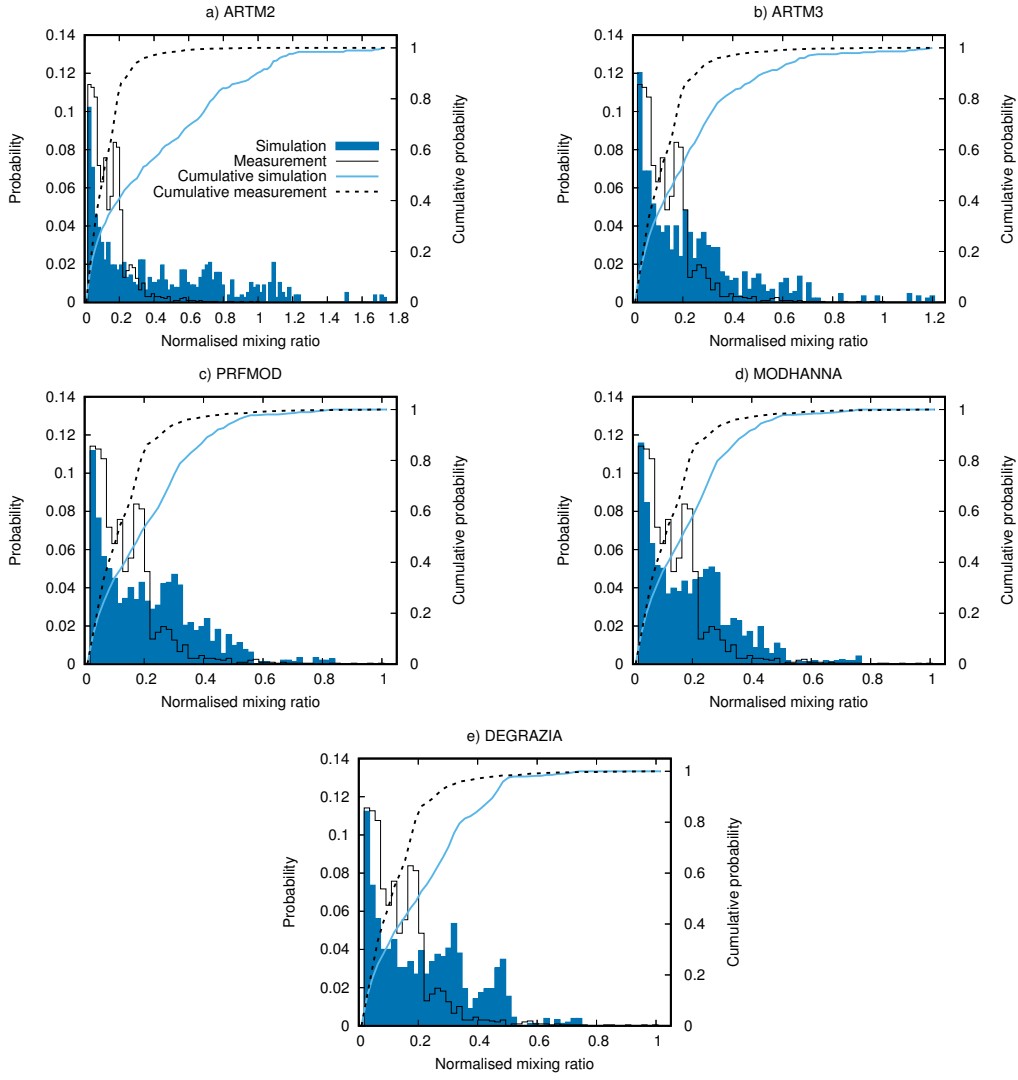

**Figure 10.** Probability distribution (bars) and cumulative probability distribution (lines) of simulated and measured mixing ratios of the five turbulence models. The PDs are normalised according to the maximum mixing ratio of the measurements and the integral of simulated and measured PDs are equal. Mixing ratio values below $1\,\text{ppmv}$ are not considered in the PDs and CPDs.

indistinct peaks of the plume not represented in the simulations. MODHANNA can be identified as the turbulence model that shows the best agreement with the observations concerning the PDs and CPDs, i.e. the occurrence of the mixing ratio values is most similar to those of the measurement. To quantify the similarity and to decide whether simulations and measurements are significantly different three statistical test were applied: The Z-test, the Kolmogorov-Smirnov (KS) test and the Cramér-von

Mises (CvM) test (Conover, 1980; Wilks, 2006; University of Oregon, 2020). The Z statistic represents the distance between



**Table 5.** Z statistics, Kolmogorov-Smirnov (KS) statistics and Cramér-von Mises (CvM) statistics of the mixing ratio distributions of the five turbulence models. The p-values are given in brackets, respectively. The significance level is 0.05.

| Turbulence model | Z statistic | KS statistic | CvM statistic |
|---|---|---|---|
| ARTM2 | 16.8 | 0.45 (0.00) | 25.1 (0.00) |
| ARTM3 | 11.5 | 0.23 (0.00) | 12.8 (0.00) |
| PRFMOD | 13.5 | 0.24 (0.00) | 15.5 (0.00) |
| MODHANNA | 10.2 | 0.20 (0.00) | 9.8 (0.00) |
| DEGRAZIA | 15.5 | 0.27 (0.00) | 19.8 (0.00) |

the means of two PDs normalised to the standard error. Statistics below 2 indicate no significant difference between the distributions. Additional interpretation limits are given in Supplement S7. The KS statistic represents the supremum of the distance between two CPDs while the CvM statistic is proportional to the integral of the distances between two CPDs. For both we assumed a significance level of 0.05. The statistics and their p-values (in brackets) are summarised in Table 5. The three statistical

tests show that all simulated mixing ratio distributions differ significantly from the observed one. Nevertheless, the statistics can be used to rank the models. MODHANNA shows the best agreement with the observations i.e. the distribution of mixing ratio values in the transects is most similar to those of the observations compared to the other turbulence models. The statistical tests rank ARTM3 second but this may be biased by the statistical tests being very sensitive to deviations in the regions of the PDs with high numbers of low mixing ratio values. We want to point out that the results do not mean that the MODHANNA

model produces mixing ratio peaks that are structured like the observed ones but the relative occurrence of mixing ratio values is most similar.

To compare the simulation results, the integral of the mixing ratio values along the flight path below the simulated boundary layer top (see Fig. 7) within the plume is shown in Table 6. We used the method above to get rid of the baseline fluctuations of the excess mixing ratios to calculate the integrals. This procedure is also applied to the simulations. Except for ARTM3,

there is a good agreement between the modelled and the measured data: The deviation is less than 13%. Concerning ARTM3, there is a strong vertical gradient in the simulated mixing ratios of the plume above $700 \, \text{m}$ as it is illustrated in the Figs. 8 b and 9. Tracers are stronger diluted (accumulated) in the upper (lower) half of the PBL than for other turbulence models. This corresponds to the findings of Sec. 4. Since the flight path is mainly located in the upper half of the PBL, the integral along the flight path results in a lower value for ARTM3. The higher mixing ratios in the lower half of the PBL might become important

when simulations are used for radiation exposure assessment. The results suggest, that the original assumption of the emission rate may not deviate much from the actual value. However, observations below $600 \, \text{m}$ are necessary to get a more complete comparison of simulated and actual plume.

The deviation between simulations of the five turbulence models and observation at a specific position can be assessed using density scatter-plots as given in Fig. 11. All mixing ratio values larger than $1 \, \text{ppm}$ along the flight path below the simulated



**Table 6.** Integrals of mixing ratio values (values below 1 ppmv are not considered) along the flight path for simulations $A_{\text{sim}}$ and observations $A_{\text{obs}}$ within the simulated PBL (see Fig. 7) given in ppmv · km as well as their ratio.

| Turbulence model | $A_{\text{obs}}$ | $A_{\text{sim}}$ | $A_{\text{sim}} A_{\text{obs}}^{-1}$ |
|---|---|---|---|
| ARTM2 | 1094 | 1186 | 108.4% |
| ARTM3 | 1094 | 742 | 67.8% |
| PRFMOD | 1094 | 1194 | 109.2% |
| MODHANNA | 1094 | 1114 | 101.8% |
| DEGRAZIA | 1094 | 1236 | 112.9% |

boundary layer top are considered. As a guide to the eye the regression with slope $m = 1$ is given as a dashed line and represents the equality of simulated and observed mixing ratios. The deviation from this equality by the factor two is confined by the red dashed dotted lines. It is not expected to find a lot of data points at the regression $m = 1$ due to the fundamental differences of the data set properties of simulation and observation. However, a large amount of data points within a deviation of a factor two decreases the uncertainty. The percentage of data points within these boarders is represented as $Fac2$ given in Fig. 11. Low $Fac2$

values can also be explained by the large number of measurement data points outside the simulated plume because they are too narrow. The smallest $Fac2$ is derived for the ARTM2 model. This coincide with the unbalanced distribution of the data points around the regression $m = 1$ with the simulation overestimating the observed mixing ratios and simultaneously simulating too narrow plumes. This is represented by the orthogonal regression of data points (black line) given in the figure with a slope above three. The residual variance $\sigma_{\text{res}}^2$ quantifies the scattering of data points. The large value for ARTM2 indicates a less compact

data point distribution. ARTM3 shows a more balanced and compacter data but still distinctly overestimates mixing ratios and underestimates plume widths. PRFMOD, MODHANNA and DEGRAZIA show similar properties with $Fac2 > 50\%$, compacter, well balanced data and less overestimated mixing ratios and underestimated plume widths, with the MODHANNA model performed slightly best for the given measurement and turbulence conditions.

## 6   Conclusions

In this work we presented an extensive evaluation of ARTM with three different elements, a sensitivity analysis, an analysis of turbulence models and a comparison with aircraft observations. Based on the sensitivity analysis, we identified the stability class to be the most important input parameter followed by the roughness length, the source height and the displacement height factor. Therefore, special care has to be taken to determine the stability class for a simulation because uncertainties of this parameter cause large uncertainties in model results. This emphasises the general disadvantage of the rather coarse stability

class concept being used to describe atmospheric turbulence. A finer classification or a continuous parameter such as the Obukhov length could be a better option but would generally require detailed measurements of turbulence parameters such as friction velocity and sensible heat flux.

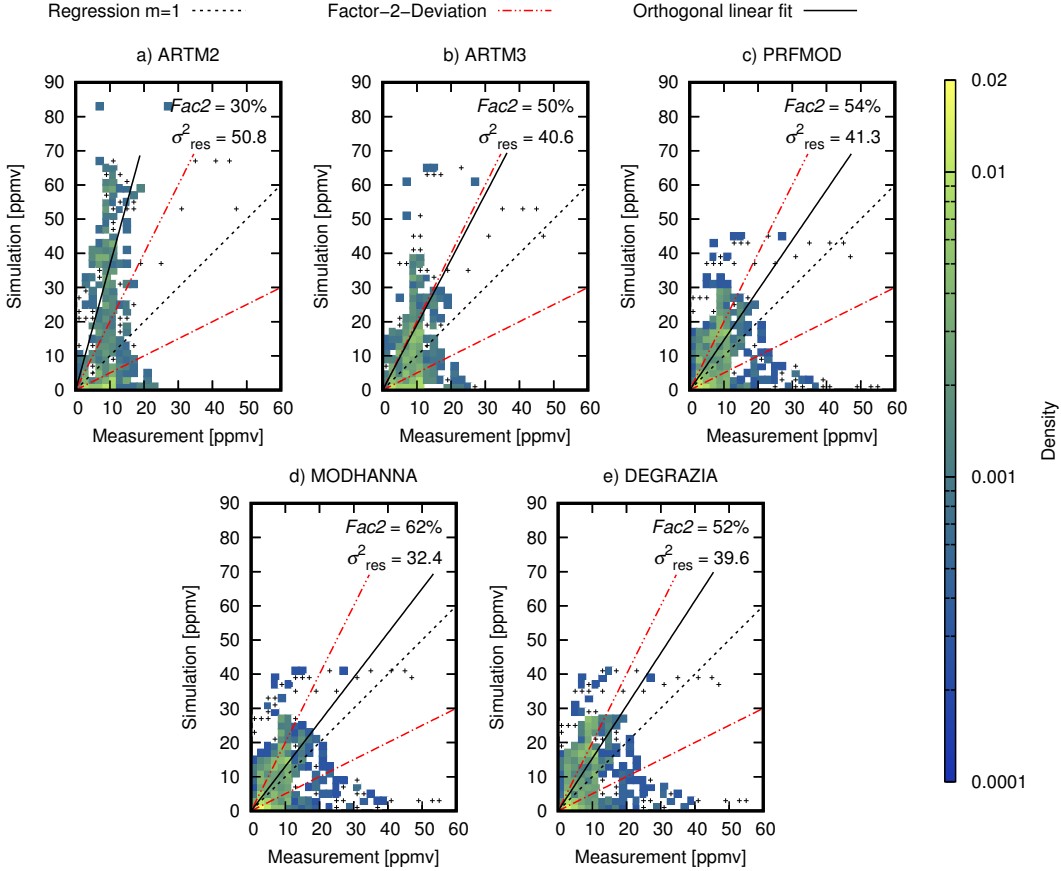

**Figure 11.** Density scatter-plots of the simulated mixing ratios of the five turbulence models ARTM2 (a), ARTM3 (b), PRFMOD (c), MODHANNA (d) and DEGRAZIA (e) against the observations. Single data points in a bin are indicated with $(+)$, more data points in a bin are colour-coded. The regression with slope $m = 1$ (dotted black line) represents the identity of simulation with measurement, the dotted dashed red line represents a deviation from the regression with $m = 1$ by the factor two and the solid black line represents the orthogonal linear fit to the data points. $Fac2$ gives the percentage of data points with deviations of not more than a factor two from the regression $m = 1$. The residual variance of the orthogonal fit is given by $\sigma^2_{\mathrm{res}}$.

In addition to the three turbulence models already implemented in ARTM 3.0.0, two further turbulence models, the MOD-
HANNA model and the DEGRAZIA model, were implemented and tested. Evaluation of the models by applying the well-
mixed condition test showed that the ARTM2, PRFMOD and MODHANNA models produced a moderate segregation of an
initially uniform concentration profile in the atmospheric boundary layer. Underestimations from the uniform concentration
occur primarily at the ground and at the top of the boundary layer with up to 10 % while overestimations occur in between
with up to 7 % at one third of the PBL. The ARTM3 model produces the strongest deviations up to four times higher than the
other models. However, near the ground ARTM3 performs best. The DEGRAZIA model showed a less inhomogeneous profile
with deviations from the uniform concentration of 5 % or less below $z = 0.8 h_{\mathrm{m}}$. The discrepancies under 6% below 80% of



the boundary layer height show good mixing properties of the planetary boundary layer for the ARTM2, PRFMOD, MOD-HANNA and DEGRAZIA model. The mixing properties of the ARTM3 model may bias simulation results when handling with $\gamma$-cloud-shine or wet deposition.

Three dimensional airborne in situ observational data measuring a power plant emission plume were compared to ARTM simulation results. The time resolution of ARTM results is one hour, which is larger than the expected time scale of the observed meandering plume and therefore, ARTM is expected to capture the time integrated real plume and not the fine structures on small scales. ARTM simulated the mean wind in agreement with the observations throughout the simulation domain. The different turbulence models simulate plume mixing ratios in the same order of magnitude as the measurements although the exact mixing ratio values depend on the emission rate. ARTM2 underestimates the plume spread under very unstable conditions and overestimates the maximum mixing ratio by a factor of two or more. The ARTM3 model produces only slightly wider plumes in lateral direction but lower maximum mixing ratio values at the upper half of the PBL. This is attributed to the inhomogeneous vertical mixing and the horizontal turbulence parametrisation of the ARTM3 turbulence model. The other turbulence models PRFMOD, MODHANNA and DEGRAZIA simulate a wider plume spread in the range of the measurements. Maximum mixing ratios are close to the measurements, the integral of the mixing ratios of simulations and observations along the flight path are comparable. The models were evaluated with measurements at heights larger than $600\,\mathrm{m}$, hence do not cover the heights below. The differences in temporal and local resolution of simulation and measurements lead to differences in the distributions of mixing ratio values. According to Fig. 10, all turbulence models underestimate the occurrence of mixing ratio values around 20% of the maximum mixing ratio of the measurements which are a result of the fine structure of the plumes. The smallest deviations in PDs and CPDs are found for MODHANNA. Using point-to-point comparisons, the ARTM2 model showed the largest deviations from the measured plume, ARTM3 shows better agreement and PRFMOD, MODHANNA and DEGRAZIA showed comparable good performances, with MODHANNA slightly matches the measurements best.

With the results of this study we showed that ARTM is able to simulate the extension and mixing ratios of a plume when the proper turbulence model is used. The ARTM3 model showed to be a suitable turbulence model for radiation exposure assessment when conservative long-term simulations are requested. However, the PRFMOD, MODHANNA and DEGRAZIA models simulate the exhaust plume closer to real exhaust plumes under the given conditions and under the limitations of temporal and spatial uncertainties. Within this validation using the in situ data from the Bełchatów power plant, the MODHANNA turbulence model performed slightly best. Further analyses with known emission terms at different atmospheric turbulence properties could lead to better validation of ARTM. The collection of measurement data in the upper and lower half of the PBL as well as transects sampling the entire extent of a plume are beneficial. Also, the use of the Obukhov length as a measure for atmospheric stability is encouraged.

*Code and data availability.* The program ARTM is available on request from the Federal Office for Radiation Protection (BfS) of Germany via artm@bfs.de. The data from the CoMeT 1.0 campaign is available at https://halo-db.pa.op.dlr.de/mission/94.



*Author contributions.* Conceptualisation: Robert Hanfland, Margit Pattantyús-Ábrahám, Dominik Brunner, Christiane Voigt; Data Collection: Alina Fiehn, Anke Roiger; Methodology: Robert Hanfland, Margit Pattantyús-Ábrahám, Dominik Brunner, Christiane Voigt; Formal Analysis: Robert Hanfland; Writing - original draft preparation: Robert Hanfland; Writing - review and editing: Margit Pattantyús-Ábrahám, Robert Hanfland, Dominik Brunner, Christiane Voigt, Alina Fiehn, Anke Roiger; Supervision: Margit Pattantyús-Ábrahám, Dominik Brunner, Christiane Voigt

*Competing interests.* The authors have no competing interests to declare that are relevant to the content of this article.

*Acknowledgements.* We thank the BfS for funding and Christopher Strobl (BfS) for his support. Furthermore, the authors would like to thank Stephan Henne (Empa) for helpful discussions.



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
