# Peer review of "The Lagrangian Atmospheric Radionuclide Transport Model (ARTM) - Sensitivity studies and evaluation using airborne measurements of power plant emissions"

_EGUsphere, 2023_

## Author Comment (AC1)

To
Editorial support team
Copernicus Publications

12 July 2023
Answer to reviewers for **egusphere-2023-245**

**Anonymous Referee #1**, 26 May 2023:

Hanfland et al. present an extensive evaluation of a Lagrangian radionuclide transport model, incorporating sensitivity analyses, tests of adherence to the well-mixed criterion, and comparisons against real-world observations downwind of power plant emissions. Overall, the analyses are solid and the presentation is fine.

I only have minor comments before suggesting the paper for publication:

**Answer to Reviewer #1:**

We thank the reviewer for her/his interest in the presented paper and his/her help to improve the quality of this manuscript.

We answer the comments below.

**Comment 1:** Line 15: reword to "allow evaliation of model performance"

**Answer to Comment 1:** Corrected.

**Comment 2:** Line 95: "Sensitiviy analysis (SA)..."

**Answer to Comment 2:** We thank the reviewer for his/her comment which results in a better understandability of the text. We added the abbreviation in the text.

**Comment 3:** Line 171: "correspond to typical"

**Answer to Comment 3:** Corrected.

**Comment 4:** Line 346: what is meant by "quite high"?

**Answer to Comment 4:** We apologize for this imprecise formulation. We replaced it by "with source heights of mainly 100 m to 200 m" in the text to make the meaning clear.

**Comment 5:** Line 386: "As tracer 14C in its bounded form as CO2 is used" doesn't seem to make sense

**Answer to Comment 5:** We corrected the formulation to "As tracer $CO_2$ with the radioactive isotope $^{14}C$ is used.". In combination with the previous sentence it makes clear that ARTM only works with predefined radionuclides that may appear in different chemical compounds.

**Comment 6:** Line 459:  "simulations' uncertainties"

**Answer to Comment 6:** Corrected.

---

## Author Comment (AC2)

To
Editorial support team
Copernicus Publications

4 October 2023
Answer to reviewers for **egusphere-2023-245**

**Anonymous Referee #2**, 22 September 2023:

GENERAL COMMENTS

This contribution is certainly of interest to the (Lagrangian, but not only) dispersion-modelling community, since, even if in application to a single model, it proposes an approach and a methodology to sensitivity studies and validation that may be adopted by other researchers.

**Answer to Reviewer #2:** We thank the reviewer for his/her positive opinion about the presented manuscript and for his/her very constructive comments that help to improve the quality of the presented study.

We answer the comments below.

**General Comment 1:**

After a sensitivity study of the input parameters for the standard version of the model ARTM (presumably, with ARTM2 turbulence parameterization: this could be specified in Section 3, then referring to Section 4 for its description), the rest of the article is dedicated to the evaluation of five different parameterization schemes for the Lagrangian turbulent variables.

**Answer to General Comment 1:** We thank the reviewer for pointing at this inexact description of the setup of the sensitivity analysis. In the revised version, we now specify the turbulence parametrisation in Section 3 and refer to Section 4 for its description as suggested by the reviewer.

**General Comment 2:**

A main concern for me is the partial use of Hanna's parameterization for the MODHANNA combination. Hanna (1982) determined different formulations for the three stratifications for both the wind velocity fluctuations (sigmas) and the Lagrangian time scales. For consistency and homogeneity, I would find it preferable to adopt for both the sigmas and the timescales the full set of Hanna's formulations, which are based on a scale analysis, surface- and

boundary-layer parameters. Picking up only the horizontal sigmas might end up being an ad-hoc adjustment based on some improvement in results, which might not be likewise effective in other cases. Hanna's formulation for the vertical sigma in the unstable case varies depending on four different ranges of the ratio between the actual height and the PBL height, yet they are quite simple.

**Answer to General Comment 2:** With the modification of the ARTM2 model we intended to analyse the shortcoming of the TKE simulated by ARTM2. As it can be seen from Fig. 2 a and c of the manuscript, the horizontal wind speed fluctuations of the ARTM2 formulation are smaller at heights above $0.3 * h_m$ compared to the other used turbulence formulations. This also results in a shortcoming of TKE as it can be seen in Fig. 2 g. Since the ARTM2 turbulence parameters, proposed by Kerschgens et al. (2000), are based on measurements we also used a measurement based formulation for the replacement of the horizontal. We chose the horizontal sigmas of the model suggested by Hanna (1982) because it is widely evaluated and its formulation is simple. In order to isolate the effects of the horizontal sigmas we did not use the Lagrangian time scales proposed by Hanna. However, we agree with the reviewer that the model proposed by Hanna (1982) is one of the turbulence models that should be used in further comparison attempts as we stated in the conclusions part of our manuscript. Also, we now state that the "ranking" of the turbulence models can not be generalized to other atmospheric conditions without further analyses.

**General Comment 3:**

The analysis in Section 5 is indeed thorough and rigorous. However, the observational data may be affected by a large uncertainty, given their origin from aircraft measurements. This is anticipated in the introductory part of the Section, but maybe it should be better addressed and somehow a 'quantitative' indication of the uncertainty of the observations could be provided. This, because the observations are then used to test and 'rank' the performance of the turbulence parameterizations, whose mutual differences might occur to lie inside the uncertainty range of the observations themselves.

**Answer to General Comment 3:** Despite the fact that the measurements were taken on an aircraft, the uncertainty is small and has little impact on our analysis. The uncertainties of the different aircraft measurements are discussed in detail by Klausner et al. (2020). We added some more information in the manuscript:

$CO_2$ was measured with a cavity ring-down spectroscopy analyser (G1301-m, Piccaro) specifically modified for the airborne deployment and with an uncertainty of $\pm 0.15$ppmv and a temporal resolution of 1s. Details of the measurement equipment and uncertainties are described by Klausner et al. (2020). The sampling repetition and the velocity of the aircraft results in a spatial resolution of 20m to 25m because during the sampling time $CO_2$ and two other substances are measured sequentially.

The measurement uncertainty is mentioned in Subsec. 5.1. Also, we included the discussion of the impact of the measurement uncertainty in the Subsection 5.5 .

**General Comment 4:**

In the conclusions, some suggestions and some findings that may be generalised are provided. Following validations of the model, in its different configurations and parameterizations, for the other atmospheric stratifications will be welcome, especially for stable conditions where turbulence parameterizations face their main challenge.

**Answer to General Comment 4:** We modified the conclusions section to make clearer that our findings are limited to unstable stratification. We also agree with the reviewer that further investigations should be made for stable and neutral conditions and added a corresponding sentence.

In the following there are some specific comments, referred to the Line Number "L XX". I think the manuscript can be considered for publication after revision.

**Comment 5:**

L 87-92. In addition to citing the differences with other models using prognostic meteo fields, it would be worth including some references of similar approaches based on a diagnostic mass-consistent model driving an LPDM

**Answer to Comment 5:** We fully agree that also models more similar to ARTM should be mentioned and therefore added references to SWIFT/micro-SWIFT and CALMET.

**Comment 6:**

L 91.  (...) 'in the vicinity': how much close?

**Answer to Comment 6:** Typical applications of this approach extend  from about 10 km to a few hundred of km (Ratto et al., 1994). Takeuchi and Adachi (1990), for example, simulated an area of 300 km x 300 km while using more than 100 anemometer positions as input for their diagnostic wind field model. In contrast to this, ARTM uses only one single anemometer position. Thus, the horizontal extent is typically chosen to be up to about 20 km depending on the terrain. Complex terrain influences the wind flow and reduces the size of the domain ARTM can work with. We now added a similar explanation to the text.

**Comment 7:**

L 95.  Maybe better to define 'SA' here instead of L 39.

**Answer to Comment 7:** We followed the reviewer's suggestion and defined SA at the suggested position.

**Comment 8:**

L 162. The top of the domain is at 300 m: was it high enough to resolve the (very)unstable conditions and their effect on the opening of the plume volume by turbulent diffusion?

**Answer to Comment 8:** The simulation domain actually had a vertical extent of 1500 m and the top layer had a thickness of 300 m. We now changed this sentence to make the description of the vertical structure clearer.

**Comment 9:**

L 169. In general, in the boundary-layer formulas, it is common to use the zero-plane displacement (say, $z_d$ in m), while here its 'factor d' is introduced: it would be worth making explicit the relationship between $z_d$ and d, which can be inferred only later by Table 4.

**Answer to Comment 9:** We thank the reviewer to point on this incomplete description of the input parameters for the SA. We now describe in more detail the connection between the zero-plane displacement and the zero-plane displacement factor and give a motivation for its usage.

**Comment 10:**

L 220. "The strong influence of hs on this target quantity is intuitively understandable, but interestingly SC is still more important."

This can be somehow expected, since the stability conditions determine the potential vertical dispersion of the plume, thus having an influence on the effectiveness of the horizontal transport and, consequently, on the location of the maximum at the ground. As known, in stable conditions, being vertical motions suppressed, the plume may travel longer distances thus maximum at the ground may be found farther from the source; whereas, in unstable stratification the downdrafts due to the thermals may bring the plume released from high sources to hit the ground relatively close to the source itself.

**Answer to Comment 10:** We agree and deleted the sentence.

**Comment 11:**

L 236. It would be worth explaining here why only the unstable formulations are presented, even if this becomes clear later. This, also in view of the fact that stable conditions are critical for their impact, given that the pollutant tends to remain inside a shallow boundary layer.

**Answer to Comment 11:** We agree with the reviewer and added a sentence making clear that the formulations are limited to unstable conditions. We also added an explanation of this limitation.

**Comment 12:**

L 238-239. "For simplicity the zero-plane displacement is not taken into account in the following equations."

Not clear to me whether this is just to simplify the formulations as reported in the manuscript or if the zero-plane displacement is not used in the formulations of the ARTM model itself.

**Answer to Comment 12:** We changed the text and made clear that ARTM uses the zero-plane displacement height in its implementation (we added a reference for this) but for the sake of simplicity we present the formulations of the turbulence variables without it.

**Comment 13:**

L 249. Since, as the authors state, the ARTM2 model is not widely used, some more information on what is based on and on how the coefficients are derived would be useful.

**Answer to Comment 13:** We thank the reviewer to point on this shortcoming of background information about the ARTM2 model. We added a description of the origin of this model and associated references.

**Comment 14:**

L 260. What are the implications of 'mixing' the ARTM2 vertical sigma with the horizontal sigmas by Hanna's model? Why not use the complete Hanna formulations, also with his Lagrangian time scales?

**Answer to Comment 14:** Please see the answer to the General Comment 2 for the implications on the used sigma components and the Lagrangian time scales. We added a motivation for the modified turbulence model.

**Comment 15:**

L 306. A dot missing after "are used"

**Answer to Comment 15:** Corrected.

**Comment 16:**

L 311. It would be worth specifying whether the profile assigned to the wind velocity is the actual one for unstable stratification; the authors may consider including a figure for it, even just in the supplementary material. Also, a wind speed of 1 m/s at 10-m height corresponds to rather low-wind conditions, is there a reason for this choice?

**Answer to Comment 16:** The wind profile is the one for unstable stratification. The wind speed of 1 m/s at a height of 10 m is close to the minimum input value of ARTM of 0.7 m/s. The performed simulations represent the limit of low wind speed. However, we now included the well-mixed condition test with a wind speed of 2.3 m/s (at 10 m height) derived from measurements under very unstable atmospheric conditions. The simulations with 2.3 m/s at the anemometer showed no significant difference in the concentration profiles compared to the low-wind condition. We now put the well-mixed condition test with low-wind speed in the Supplement.

**Comment 17:**

L 394-395 Wind-meandering is generally associated with non-turbulent oscillations of the horizontal wind velocity, so in principle it cannot be expected to be represented in the turbulence spectrum, therefore to be resolved by turbulence parameterisations. The authors might comment on this aspect.

**Answer on Comment 17:** We agree that the observed meandering was not necessarily associated with turbulence, but this is very difficult to judge from the limited observations. The transition from turbulent to non-turbulent motion is not a sharp one and turbulent Eddies can be quite large at this altitude. Since the ARTM results represent a 1-hour average, we expect them to represent the dispersion by all processed acting on timescales smaller than 1 hour while motions on larger time scales are represented by the changes of the mean wind direction.

**Comment 18:**

L 449-450: I am not sure how to interpret the sentence:

"However, in Fig. 7 the highest maximum mixing ratios at wall 1 occur at different transects for the simulations and the observation"

Should it be Fig. 8 instead of Fig. 7? Wall 1 corresponds to transect 2 (Fig. 6), while looking at Fig. 7 the maximum mixing ratios are found, for both simulated and observed data, in transect 1: is this the meaning?

**Answer on Comment 18:** We apologize that the sentence was not sufficiently clear. We wanted to comment on the differences of the two lowest peaks in wall 1 between simulations and observation. We rephrased the sentence to make the meaning clear.

**Comment 19:**

L 532-533: "The mixing properties of the ARTM3 model may bias simulation results when handling with γ-cloud-shine or wet deposition."

This sentence sounds a bit out of (this) context: consider explaining and justifying it more, or removing it.

**Answer to Comment 19:** We removed the sentence as it is indeed out of the context at this position.

**Comment 20:**

L 553-561. The authors might find of interest some other comparisons between Hanna and Degrazia parameterizations, in the following papers:

Carvalho et al., 2002. Atmospheric Environment, 36, n. 7, 1147-1161

Trini Castelli et al. 2014. Quart J Roy Meteorol Soc., 140, 2023-2036

Trini Castelli et al. 2014. In: Steyn DG, Builtjes P (eds) Air pollution modeling and its application XXII. Springer, Berlin, 529–534

**Answer on Comment 20:** We thank the reviewer to refer to the mentioned publications. They are of certain interest when comparing the Hanna and Degrazia models with the used observations. This will be very interesting to study in future attempts. We added a sentence to the conclusions section, respectively.

**References:**

Hanna, S. R. (1982). Applications in Air Pollution Modeling. In F. T. M. Nieuwstadt & H. van Dop (Eds.), *Atmospheric Turbulence and Air Pollution Modelling: A Course held in The Hague, 21–25 September, 1981* (pp. 275-310). Springer Netherlands. https://doi.org/10.1007/978-94-010-9112-1_7

Kerschgens, M. J., Nölle, C., & Martens, R. (2000). Comments on turbulence parameters for the calculation of dispersion in the atmospheric boundary layer. *Meteorologische Zeitschrift*, *9*(3), 155-163. https://doi.org/10.1127/metz/9/2000/155

Klausner, T., Mertens, M., Huntrieser, H., Gałkowski, M., Kuhlmann, G., Baumann, R., Fiehn, A., Jöckel, P., Pühl, M., & Roiger, A. (2020). Urban Greenhouse Gas Emissions from the Berlin Area: A Case Study Using Airborne CO2 and CH4 In Situ Observations in Summer 2018. *Elementa Science of the Anthropocene*, *8*(1). https://doi.org/10.1525/journal.elementa.411

Ratto, C. F., Festa, R., Romeo, C., Frumento, O. A., & Galluzzi, M. (1994). Mass-consistent models for wind fields over complex terrain: The state of the art. *Environ. Softw.*, *9*(4), 247-268. https://doi.org/10.1016/0266-9838(94)90023-X

Takeuchi, Y., & Adachi, T. (1990). Seasonal variations of atmospheric trajectories arriving at Osaka in the Kansai area. *Atmos Environ A-Gen*, *24*(8), 2011-2018. https://doi.org/https://doi.org/10.1016/0960-1686(90)90235-F